# Gender differentials in the timing and prognostic factors of pubarche in Nigeria

**Adeniyi Francis Fagbamigbe**[1]*, **Mary Obiyan**[2], **Olufunmilayo I. Fawole**[1]

**1** Department of Epidemiology and Medical Statistics, Faculty of Public Health, College of Medicine, University of Ibadan, Ibadan, Nigeria, **2** Department of Demography and Social Statistics, Obafemi Awolowo University, Ile-Ife, Nigeria

* franstel74@yahoo.com

**Data Availability Statement:** The link to the data is https://www.kaggle.com/datasets/adeniyifagbamigbe/purbache-aff.

**Funding:** The author(s) received no specific funding for this work.

## Abstract

Paucity of data exists on the timing of puberty, particularly the pubarche, in developing countries, which has hitherto limited the knowledge of the timing of pubarche, and assistance offered by physicians to anxious young people in Nigeria. Factors associated with the timings of puberty and pubarche are not well documented in Nigeria. We hypothesized that the timing of pubarche in Nigeria differs by geographical regions and other characteristics. We assessed the timing of pubarche among adolescents and young adults in Nigeria and identified prognostic factors of the timing by obtaining information on youths' sexual and reproductive developments in a population survey among in-school and out-of-school youths aged 15 to 24 years in Nigeria. A total of 1174 boys and 1004 girls provided valid information on pubarche. Results of time-to-event analysis of the data showed that mean age at pubarche among males aged 15 to 19 years and 20 to 24 years was 13.5 ($SD$ = 1.63 years) and 14.2 ($SD$ = 2.18 years) (respectively) compared with 13.0 ($SD$ = 1.57 years) and 13.5 ($SD$ = 2.06 years) among females of the same age. Median time to pubarche was 14 (Interquartile range (IQR) = 3) years and 13 (IQR = 3) years among the males and females, respectively. Cumulatively, 37% of the males had attained pubarche by age 13 years versus 53% among females, 57% vs 72% at age 14, and 73% vs 81% at age 15. The likelihood of pubarche among males was delayed by 5% compared with females (Time Ratio (TR) = 1.05: 95% CI = 1.03–1.05). Every additional one-year in the ages of both males and females increases the risk of pubarche by 1%. Similar to the females, males residents in Northeast (aTR = 1.14, 95% CI: 1.07–1.21), in the Northwest (aTR = 1.20, 95% CI: 1.13–1.27) and in the Southwest (aTR = 1.18, 95% CI: 1.11–1.26) had delayed pubarche than males from the South East. Yoruba males had delayed pubarche than Ibo males (aTR = 1.06, 95% CI: 1.01–1.12). Age at pubarche among adolescents and young adults in Nigeria differed among males and females with earlier onset among females. Pubarche timing varied mainly by ethnicity, region, and location of residence. Our findings will aid medical practitioners in providing appropriate advice and support on pubarche-related issues among adolescents in Nigeria as it could help douse pubarche anxiousness in relation to request for medical assistance.

**Competing interests:** The authors have declared that no competing interests exist.

## Introduction

Puberty has been reported to be a dynamic period which characteristically encompasses physical growth, maturation of sexual abilities and psychosocial achievement that mostly starts when a child is aged 8 to14 years [1, 2]. Pubarche, the development of pubic hair, is one of the characteristics of puberty [1, 3]. However, the rate of physical development varies widely during adolescence. The variations sometimes become a source of excitement, difficulty and discomfort for youths, because some develop more slowly than their peers. As a result, adolescents with slower developments often feel self-conscious about their bodies' lack of maturity, relative to their peers [4]. As important as the timing of purbache is, Many adolescents transition to adulthood without correct and adequate information on what puberty entails and how it affects their social and cognitive behaviours. Gender difference in the timing of puberty among boys and girls (10–14 years for girls and 12–16 years in boys) have been identified [3]. Adolescent girls in particular often experience frustration, confusion, agitation, disappointments and resentfulness as they do not receive the same kind of attention as their more physically mature friends [5]. The current study is aimed at assessing the timing of pubarche among boys and girls in Nigeria as well as identifying background, biological and environmental/societal characteristics affecting the timing.

Boswell et al. identified the need to understand the timing of pubarche as it enhances psychosocial maturation and reproductive capacity [1]. Although different studies have estimated the timing of pubarche and identified its associated risk factors in most regions of the world [6–10], such studies are scarce for the Nigeria setting. Besides, the available ones are outdated [11, 12]. The need to update knowledge and to inform surrounding adults to ease anxieties that often comes with the unpredictability of age at pubarche [1, 5] necessitated this study. Instances of premature pubarche, before attainment of age 8 in girls and age 9 in boys have been reported [2]. Literature has illustrated the validity of the Tanner 5 staging models in pubic hair development among both boys and girls [13, 14].

Different timings of pubarche have been reported in the literature [6–10]. A cross-sectional study of sexual maturation of healthy Chinese girls showed that the average ages of onset of pubic hair development Tanner Stage 2, which is characterised with a sparse growth of long, slightly pigmented hair, straight or curled, at base of the penis or along labia, were 11.16 (95% CI: 11.03–11.29) years and 12.40 (95% CI: 12.25–12.55) years for Tanner Stage 3 [7]. In the same vein, a South African study found average age at the initiation of puberty partly assessed by age at pubic hair development as 9.8 to 10.5 years. The authors reported that age at the onset of puberty has been constant for 10 to 15 years preceding the study. They, however, noted a decline of about one year in the onset of pubarche among boys over ten years [6].

The differences in timings of onset of pubarche have been ascribed to biological and sociodemographic variabilities including ethnicity, geographical location, sex, health status, genetics, nutrition, and activity level [1, 8, 15]. The knowledge of risk factors associated with pubarche could help medical practitioners provide evidence-based care and guidance to adolescents and their guardians. Several studies have reported socioeconomic characteristics to be significantly associated with the early timing of puberty among boys and girls; however, little is known about the association between educational attainment, incidence and timing of pubarche [3, 16–18]. However, an earlier Nigerian study has established an association between socioeconomic status, education and health outcomes [19]. We hypothesized that better educational attainment could improve decisions on healthy food that may, in turn, affect the onset of pubarche; pubertal timing has been demonstrated to be a function of nutritional intake during childhood [20]. Nutritional intakes have been reported to be different among rural and urban children, since both have differential access to different types of foods [15].

Literature has confirmed that nutritional status is an important physiological regulator of puberty-related developments among adolescents [20–22]. Also, earlier studies found childhood family-related adversity experiences to be associated to earlier age at pubarche and menarche which has correlated with the likelihood of CVD in adulthood [9, 10].

The association between place of residence and timing of pubarche can be explained by the differentials in nutritional intake in rural and urban areas [20]. An earlier Nigeria study also noted that rural-urban differences in pubertal development among school adolescents with earlier attainments of pubarche among urban adolescents [8]. Also, an India study reported rural-urban differences in age at pubarche [23].

Race and ethnicity have been identified as associated factors with age at pubarche [24]. This position was ascribed to differences in the levels of metabolism and insulin across different races. Nigeria is a multi-ethnic country Southwest Nigeria is dominated by the Yorubas, the Northeast and Northwest regions by the Hausa/Fulanis, and the Southeast by the Ibos. Other regions have mixed ethnic concentrations. In particular, we included the region as an explanatory variable in this study because Nigeria is divided into six geographical regions based mainly on distinct geographical locations and differed in culture, language, social-economic capabilities, types of food, diet etc. Similar assertion has been made in the literature [25]. Besides this brief introduction, we presented the theoretical/conceptual framework for the study, statement of the problem and purpose of the study, significance of the study, methodology consisting of data sources and statistical analysis procedures, the result interpretation, discussions followed by strengths and limitations, the recommendations, and conclusions.

## Theoretical / conceptual framework

The current study is based on a theoretical and conceptual framework of the timing of pubarche identified in the literature. We adapted the ecological model which suggested a complex interplay among individuals, relationships, community and societal factors affecting the timing of pubarche [26]. The model showed the diverse and complex range of characteristics of adolescent development that are more rooted in culture than in human biology or cognitive structures [4]. The theory is based on the fact that the influence of parental and peer relationships, as well as the broader culture, shapes many aspects of adolescent development. Also, Lumen Learning et al. stated that culture is learned and socially shared, and it affects all aspects of an individual's life [4]. Also, social responsibilities, sexual expression, and belief-system development may vary based on culture. Furthermore, many distinguishing characteristics of an individual are all societal- and culturally-based [4]. Based on the identified factors in literature, we developed a conceptual framework to identify factors associated with timing of pubarche in Nigeria. The conceptual framework suggested a pathway of association between individual characteristics (sex and age) as the main determinate variables, household characteristics (wealth status) and other factors such as the rural-urban location of residence, educational attainment, and ethnicity as well as the geographical place of residence and the onset of pubarche (the society/culture).

## Statement of the problem and purpose

In Nigeria, as in most developing countries, there is a paucity of current data regarding the timing of puberty, and pubarche, unlike in the more robust literature in developed countries. For instance, the most recent study on timing of puberty development in Nigeria [27]. The study did not cover pubarche development. Other related studies were on knowledge and perception of parents and guardians on pubertal developments among youths [8, 28]. The non-availability of up-to-date data that may guide stakeholders has made it difficult for young

children and their guardians to access reference information. More so, reproductive health education is yet to be incorporated into the school curriculum in Nigeria. The current study will bridge this gap by updating the body of knowledge on age at onset of pubarche among both boys and girls in Nigeria. Although the onset of pubarche has been studied in other countries, the situation in Nigeria may be different considering the differences in socioeconomic situation, geographical location, cultural practices and ethnicity, hence the need for the current study. Thus, we hypothesized that the timing of pubarche in Nigeria differs by geographical divisions and other characteristics and that age at pubarche in the Nigeria has changed over time compared with what is available in literature.

The purpose of the study is to estimate the timing of the onset of pubarche among boys and girls in Nigeria and identify factors that are associated with its timing. The outcomes of our study can be used to reassure adolescents who may be undergoing delayed puberty that they are probably not isolated. Since puberty brings about critical social change and exposes children to new peers with different life experiences, practices, and behavioural patterns, the outcome of this study will help parents and guardians to prepare for this critical stage of life.

## Significance of the study

The significance of the current study lies in its contribution to the understanding of how socio-cultural differences in Nigeria could impact the timing of the onset of pubarche in the country. The peculiarities are in terms of cultural practices, food types, socioeconomic situation, geographical location, and ethnicity make this study necessary. The results of this study can be used to correct wrong impressions regarding pubarche and inform the general society on the normal variations in the range of onset of pubarche and when it may become necessary to seek medical help. Recent reports showed that the lack of information on puberty may be detrimental to the reproductive health of the teenagers and youths [4]. As oftentimes uninformed adolescents engaged in sexual risks behaviour which could lead to unwanted pregnancy and sexually transmitted diseases [28].

## Methods

### Study area

The study was carried out in Nigeria, a country with an estimated population of 190 million, of which 19.6% are adolescents and young adults aged 15 to 24 years and that had an annual growth rate of 2.4% in 2017 [29]. Administratively, Nigeria is made up of 36 states and a Federal Capital Territory. It is divided into six geopolitical regions: North West, North East, North Central, South East, South West, and South-South.

### Study design

The data used for this analysis was collected during the "Youths and adolescents sexual and reproductive health survey" in 2017. The study design was a cross-sectional nationally representative population survey during which the participants recalled past events. Primary data were obtained by the authors from the survey. The participants were male and female adolescents and young adults in Nigeria. The participants were randomly drawn from households in rural and urban areas using the updated master sample frame of rural and urban localities and Enumeration Areas (EA) developed by the National Population Commission [29].

## Sampling technique

Probability sampling technique using a multi-stage cluster sampling method was used to select eligible respondents. Stage 1 involved the random selection of one state from each of the regions in Nigeria. In Stage 2, one rural and one urban local government area (LGA) each was randomly selected using stratified random sampling from the selected states in Nigeria. In Stage 3, 15 enumeration areas (EA) within the selected rural and urban LGAs were selected randomly. In Stage 4, 16 households within the EAs were selected randomly while Stage 5 involved the selection of one youth per household for interview. Four hundred and eighty (480) respondents were sampled for the survey in each of the 6 states. A total of 2952 participants took part in the survey.

## Data collection

A pre-tested semi-structured interviewer-administered questionnaire was used in June 2017 in a study on the sexual and reproductive health of adolescents and young adults in Nigeria. The interviewers were duly trained before data collection and supervised closely in the field. The questionnaires consisted of sections that focussed on contemporary issues on youths' sexuality, reproductive health, childbirth, marriage, HIV/AIDS transmission, and prevention knowledge. The respondents were asked: How old were you when you first notice hair growth in your pubic areas? Among the 2952 participants, a total of 1174 males and 1004 females provided valid responses to age at pubarche. We excluded respondents who did not provide a valid response to the question, respondents who couldn't remember their age as of the time of first pubic hair growth and others who provided ages that are not consistent with other responses. We accounted for recall bias by first asking: How old (age years) were you when you first notice hair growth in your pubic areas? Then we asked cross-validation questions on other pubertal outcomes such as age at menarche and age at puberty. The responses to these questions were used to ascertain consistencies of the reported ages at pubarche. This minimised errors that could be associated to recall bias. Inconsistent responses were clarified with the respondent, while those that couldn't be clarified were excluded from further analysis. All analyses in this study were based on data of those who provided a valid response.

## Statistical analysis

The outcome variable is pubarche, the age at onset of pubic hair growth. The respondents were asked: At what age did you first notice hair growth in your pubic? The responses to this question were in years because (1) using months will lead to higher inaccuracies as the respondents had to recall their ages in months (2) year has been used widely as a valid metric for a similar retrospective report of age at pubarche [1, 8]. The independent variables are based on our conceptual framework that individual characteristics, household characteristics as well as the geographical place of residence influences onset of pubarche. The independent variables considered in this study are the location of residence, region, educational attainment, wealth status and ethnicity as used in earlier studies [1, 8, 15]. Descriptive statistics including percentages, rates, [proportions and confidence intervals were used to describe the characteristics of the respondents. STATA V16 and IBM SPSS V25.0 software were used to carry out the survival analysis of the timings of pubarche among the respondents and also to identify factors influencing it. Statistical significance of all tests was determined at $p = 0.05$.

## Analysis of time to event (survival analysis)

The technique, otherwise known as the history of the event is a statistical procedure that monitors time duration until one or more events of interest happen. Two quantitative terms are

paramount in survival analysis. They are the survivor function *S(t)* and hazard function *h(t)*. For this study, the survivor function gives the probability that a child "survives" longer than some specified time *t* without developing pubic hair, while the hazard function estimates the instantaneous chance per unit time to develop pubic hair after time *t*, given that the child has not developed pubic hair before that time. In essence, *h(t)* is the failure rate at time *t* among those children who have not developed pubic hair *t*. The survival and hazard functions are estimated respectively by

$$s(t) = S'(t) = \frac{d}{dt}S(t) = \frac{d}{dt}\int_t^\infty f(u)du = \frac{d}{dt}[1 - F(t)] = -f(t) \tag{1}$$

and

$$\lambda(t) = \lim_{dt \to 0} \frac{\Pr(t \le T < t + dt)}{dt. \, S(t)} = \frac{f(t)}{S(t)} = \frac{S\prime(t)}{S(t)} \tag{2}$$

Survival analysis requires two variables to be explicitly determined. They are the survival time and the censoring index. The survival time or follow-up time is assumed to be a discrete random variable that takes on only positive integer. In the current study, the population at risk are all male and female participants since they all have tendencies to develop pubic hair at one time or the other. For those who have developed pubic hair as of the day of the survey, the survival time is the age at which they developed pubic hair. Participants who were yet to have pubic hair on the day of the survey were consequently censored and their current age recorded as the censoring time.

Kaplan-Meier estimates were used to compute the survival functions at time *t* where $n_j$ is the number of participants who are at the risk of developing pubic hair at time $t_j$ and d*j* is the number of subjects that developed pubic hair at time $t_j$. We fitted Accelerated Failure Time (AFT) models consisting of exponential, Weibull, Gompertz, lognormal, log-logistic, and generalized gamma models to the data. The STREG package in STATA was used to estimate the maximum likelihood estimation for the parametric regression survival-time models. The accelerated failure time implies a deceleration of time or, in other terms, an increment in the expected waiting time for an event of interest to occur. The AIC, BIC and *-2loglikelihood* was used to choose the best model for the data. The generalised gamma model (GGM) consistently had the best performance for the three criteria used. The time ratio (TR) is the output for comparison in AFT models. It is a ratio of times to pubarche among participants in two different categories of a variable. TR>1 implies an earlier experience among those been compared to the reference group, TR<1 implies a delayed experience among those been compared to the reference group, whereas TR = 1 implies no difference. The adjusted time ratio (aTR) are the corresponding estimates from the multivariable models.

The model assume that treatments have a multiplicative effect on survival time that is consistent over time. It does not assume constant hazards but accounts for the effects of the covariates directly on survival times rather than the hazard rates as in the proportional hazard (PH) model [30]. The Juneau et al.'s diagnostics procedures for checking the gamma distribution assumption in generalized linear models were used [31]. In particular, we evaluated the validity of the assumption, quantile-quantile (QQ) plots and checked whether estimated scale and shape parameters differed significantly between males and females [31, 32]. This model is also best suited to manage inherent frailty which is very likely due to the sampling strategies used in this study.

**Ethical considerations.** Ethical approvals for the study were obtained Obafemi Awolowo University Ethical Board (IPHOAU/12/582). A written inform consent was obtained from the

study participants. The collection procedure posed no harm to the participants. Rather, the information gathered were used to benefit them and the country at large. Data were collected anonymously, held confidentially and feedback was given to participating communities. Furthermore, the research team obtained permissions from the community leaders, village or area chiefs in all the communities where data was collected. Written informed consent for participation in the study was obtained from all the participants aged 18–24 years and additional written consent was obtained from parents and guardians of participants who have not attained aged 18 years.

## Results

Table 1 provides information for males in the sample. Among the 1178 males, 58% were aged 15 to 19 years while 42% were aged 20 to 24 years. Most (72%) of the males had up to

**Table 1. Distribution of time to pubarche and incidence rate among males by selected characteristics.**

| Characteristics | [a]Percent (n = 1178) | Mean | Median | Incidence Rate | Survival time at | | |
|---|---|---|---|---|---|---|---|
| | | | | | 25% | 50% | 75% |
| Age (years) | | | | | | | |
| 15–19 | 58.4 | 13.5 | 14 | 0.063 | 13 | 14 | 15 |
| 20–24 | 41.6 | 14.2 | 14 | 0.058 | 13 | 15 | 16 |
| Region | | | | | | | |
| North Central | 17.4 | 13.2 | 13 | 0.067 | 12 | 13 | 15 |
| North East | 20.2 | 14.7 | 15 | 0.065 | 14 | 15 | 16 |
| North West | 10.9 | 15.2 | 15 | 0.053 | 15 | 16 | 17 |
| South East | 17.0 | 12.9 | 13 | 0.066 | 12 | 13 | 14 |
| South South | 22.4 | 13.6 | 14 | 0.073 | 13 | 14 | 15 |
| South West | 12.1 | 13.8 | 14 | 0.029 | 14 | CBC | CBC |
| Residence | | | | | | | |
| Urban | 49.8 | 14.0 | 14 | 0.057 | 13 | 14 | 16 |
| Rural | 50.2 | 13.7 | 14 | 0.064 | 13 | 14 | 15 |
| Education | | | | | | | |
| None | 4.5 | 13.8 | 14 | 0.060 | 13 | 14 | 15 |
| Pry/Quranic | 12.0 | 14.0 | 14 | 0.063 | 13 | 14 | 15 |
| Secondary | 72.2 | 13.7 | 14 | 0.061 | 13 | 14 | 16 |
| Higher | 11.4 | 14.3 | 15 | 0.059 | 13 | 15 | 16 |
| Ethnicity | | | | | | | |
| Hausa/Fulani | 34.0 | 14.6 | 15 | 0.060 | 14 | 15 | 16 |
| Yoruba | 14.3 | 13.9 | 14 | 0.037 | 14 | 15 | CBC |
| Ibo | 18.3 | 13.1 | 13 | 0.066 | 12 | 13 | 15 |
| Others* | 32.7 | 13.5 | 14 | 0.071 | 13 | 14 | 15 |
| Wealth Category | | | | | | | |
| Lowest | 34.0 | 13.9 | 14 | 0.062 | 13 | 14 | 16 |
| Average | 32.9 | 13.7 | 14 | 0.061 | 13 | 14 | 15 |
| Highest | 33.2 | 13.9 | 14 | 0.059 | 13 | 14 | 16 |
| Total | 100.0 | 13.8 | 14 | 0.061 | 13 | 14 | 16 |

[a]The median time

[b]LWSP Living with Sexual Partner

CBC = Can't be computed (<50% had experienced pubarche)

*Include Isan, Tiv, Kanuri

secondary education, and about 50% lived in the urban areas. The mean age at pubarche among males aged 15 to 19 years was 13.5 (SD = 2.3) years and 14.2 (SD = 2.6) years among those aged 20 to 24 years. The median age at pubarche age for both 15 to 19 years and 20 to 24 years respondents was 14 (Inter-quartile range (IQR) = 3) years. The mean age among males that had already attained pubarche was 13.5 years for those aged 15 to 19 years and 14.2 years for those aged 20 to 24 years. The incidence rate of pubarche, that is the probability of beginning pubic hair growth in a year, among the males was 0.061. The incidence was lower in the Southwest region at 0.029 compared with other regions. The incidence ranged from 0.053 to 0.073 in the other regions. The 25th percentile survival time among the males was 13 years, 50th percentile (the median survival time) was 14 years and 75th percentile was 16 years (Table 1).

Table 2 provides information for females in the sample. Almost two thirds (63%) of the 1004 female respondents were aged 15 to 19 years. About half (54%) lived in urban areas. The

**Table 2. Distribution of time to pubarche and incidence rate among females by selected characteristics.**

| Characteristics | Percent (n = 1004) | Mean | Median | Incidence Rate | Survival time | | |
|---|---|---|---|---|---|---|---|
| | | | | | 25% | [a]50% | 75% |
| Age | | | | | | | |
| 15–19 | 63,3 | 13.0 | 13 | 0.068 | 12 | 13 | 14 |
| 20–24 | 36,7 | 13.5 | 13 | 0.058 | 12 | 14 | 16 |
| Region | | | | | | | |
| North Central | 17,2 | 13.2 | 13 | 0.059 | 13 | 14 | 15 |
| North East | 12,7 | 13.9 | 14 | 0.072 | 13 | 14 | 15 |
| North West | 12,7 | 13.3 | 13 | 0.038 | 13 | 15 | CBC |
| South East | 18,2 | 12.0 | 12 | 0.076 | 12 | 12 | 13 |
| South South | 21,7 | 13.2 | 13 | 0.075 | 12 | 13 | 14 |
| South West | 17,3 | 13.6 | 13 | 0.061 | 13 | 13 | 16 |
| Residence | | | | | | | |
| Urban | 53,6 | 13.2 | 13 | 0.067 | 12 | 14 | 15 |
| Rural | 46,4 | 13.0 | 13 | 0.060 | 12 | 13 | 15 |
| Education | | | | | | | |
| None | 4,0 | 12.7 | 13 | 0.045 | 12 | 14 | CBC |
| Pry/Quranic | 14,8 | 12.9 | 13 | 0.057 | 12 | 13 | 15 |
| Secondary | 69,7 | 13.2 | 13 | 0.067 | 12 | 13 | 15 |
| Higher | 11,5 | 13.3 | 13 | 0.062 | 12 | 14 | 15 |
| Ethnicity | | | | | | | |
| Hausa/Fulani | 26,8 | 13.7 | 14 | 0.056 | 13 | 14 | 16 |
| Yoruba | 18,0 | 13.3 | 13 | 0.057 | 12 | 13 | 17 |
| Ibo | 21,3 | 12.3 | 12 | 0.070 | 12 | 12 | 14 |
| Others* | 32,7 | 13.3 | 13 | 0.070 | 12 | 13 | 14 |
| Wealth Category | | | | | | | |
| Lowest | 33,4 | 13.0 | 13 | 0.066 | 12 | 13 | 15 |
| Average | 33,0 | 13.3 | 13 | 0.066 | 12 | 13 | 15 |
| Highest | 33,7 | 13.2 | 13 | 0.060 | 12 | 13 | 15 |
| Total | 100.0 | 13.2 | 13 | 0.064 | 12 | 13 | 15 |

[a]The median time

[b]LWSP Living With Sexual Partner

CBC Can't Be Computed (<50% had experienced pubarche)

*Include Isan, Tiv, Kanuri

**Table 3. Percentage distribution of developing pubic hair at a given age among those that had already started and among all respondents.**

| Age (x) | Percentage among those that had pubic hair | | Percentage among all respondents | | | |
|---|---|---|---|---|---|---|
| | | | Male (n = 1178) | | Female (n = 1004) | |
| | Male (n = 1035) | Female (n = 887) | At Exact Age x | Up to Age x | At Exact Age x | Up to Age x |
| 8 | 0.29 | 0.90 | 0.25 | 0.25 | 0.80 | 0.80 |
| 9 | 0.19 | 0.68 | 0.17 | 0.42 | 0.60 | 1.40 |
| 10 | 4.06 | 4.96 | 3.57 | 3.99 | 4.38 | 5.78 |
| 11 | 5.60 | 5.75 | 4.92 | 8.91 | 5.08 | 10.86 |
| 12 | 13.14 | 22.10 | 11.54 | 20.45 | 19.52 | 30.38 |
| 13 | 18.84 | 26.16 | 16.55 | 37.00 | 23.11 | 53.49 |
| 14 | 22.80 | 20.86 | 20.03 | 57.03 | 18.43 | 71.92 |
| 15 | 17.78 | 10.60 | 15.62 | 72.65 | 9.36 | 81.28 |
| 16 | 10.14 | 3.61 | 8.91 | 81.56 | 3.19 | 84.47 |
| 17 | 3.67 | 2.14 | 3.23 | 84.79 | 1.89 | 86.36 |
| 18 | 2.71 | 1.69 | 2.38 | 87.17 | 1.49 | 87.85 |
| 19 | 0.39 | 0.34 | 0.34 | 87.51 | 0.30 | 88.15 |
| 20 | 0.11 | 0.23 | 0.09 | 87.60 | 0.20 | 88.35 |
| 21 | 0.11 | 0.00 | 0.09 | 87.69 | 0.00 | 88.35 |
| 22 | 0.19 | 0.00 | 0.17 | 87.86 | 0.00 | 88.35 |
| Not yet | - | - | 12.14 | - | 11.65 | - |

mean age at pubarche among females aged 15 to 19 years was 13.0 (*SD* = 1.6) years and 13.5 (*SD* = 2.1) years among those aged 20 to 24 years. The overall mean age among females in each of the age groups that had already attained pubarche was 13.0 for those aged 15 to 19 years and 13.5 years for those aged 20 to 24 years. The incidence rate of pubarche among the females was 0.064. The incidence was lower at 0.038 in the Northwest region compared with other regions that ranged from 0.059 to 0.076. The 25[th] percentile survival time among girls in all age groups was 12 years, 50[th] percentile (the median survival time) was 13 years while the 75[th] percentile was 15 years.

As seen in Table 3, pubarche peaked at 12 to 16 years. Among the male respondents, 13% attained pubarche at Age 12, 19% at 13, 23% at 14, 18% at 15 and 10% at 16, compared with 22%, 26%, 21%, 21% and 4% among the females at the same ages, respectively. Cumulatively, 37% of the males had attained puberty by age 13 versus 53% among females, 57% vs 72% at age 14 and 73% vs 81% at age 15 (Table 3).

As shown in Fig 1, at the respective ages, the survival probabilities of pubarche were lower among females than among the males. However, the log-rank test used in testing for equality of survival functions showed that the survivorship among males and females were significantly different.

The probability of not attaining pubarche as the participants grow older viz-a-viz their age group, the location of residence, region, and ethnicity are presented in Figs 2 and 3 for males and females respectively. The probability of not attaining pubarche with respect to respondents' age, ethnicity, location and geographical region of residence was significantly different among males while respondents' age, ethnicity, the geographical region of residence, educational attainment and were significantly different among the females. The proportion of respondents who have not attained pubarche was more evident across the region of residence of the respondents. About 50% of males in the South-West region, over 40% of females in the North-West region and nearly 40% of Yoruba ethnic males have not yet experienced pubarche.

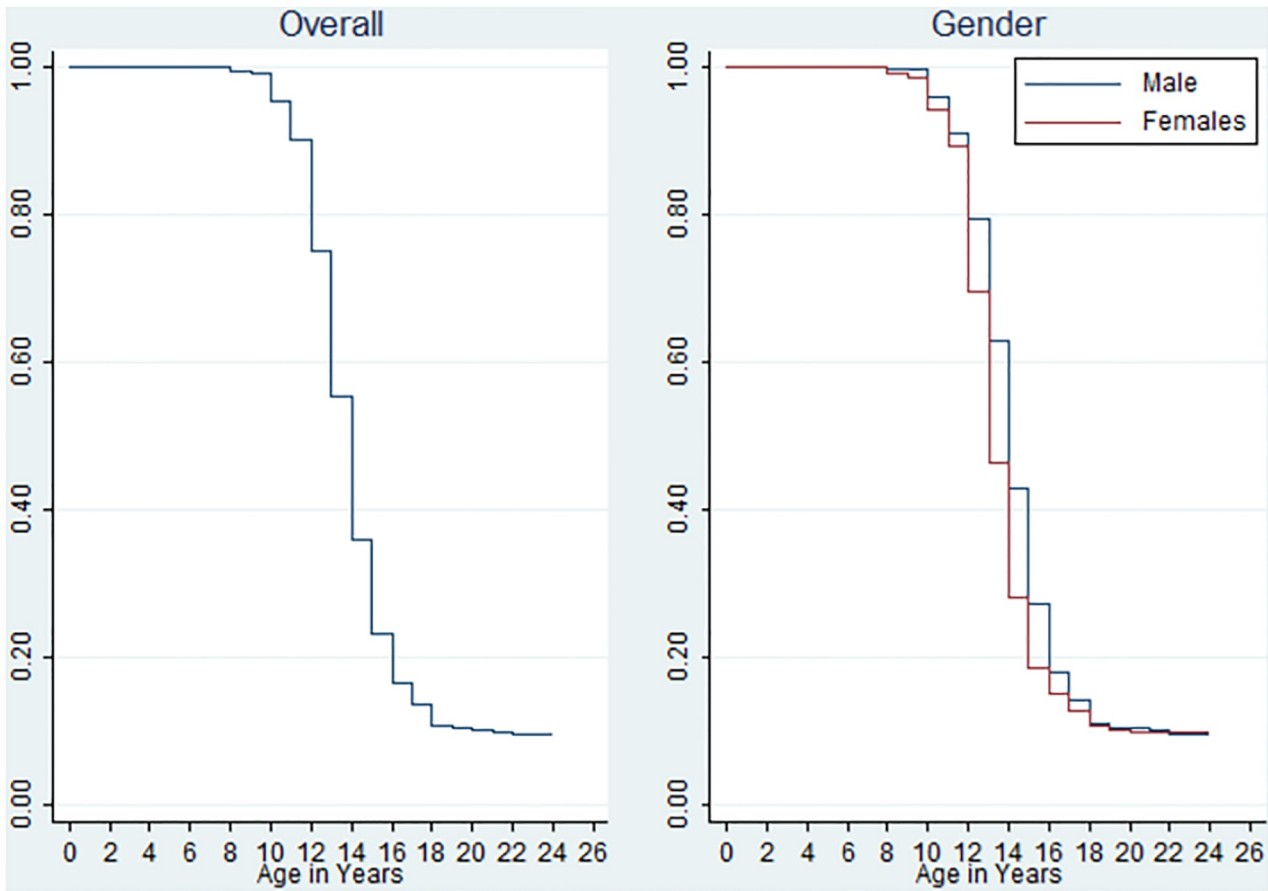

**Fig 1. Survivorship curves showing probabilities of developing pubic hair at different ages by sex.**

Table 4 shows the unadjusted and adjusted determinants for risk of pubarche. The multiple regression run separately with the timing of pubarche in the unadjusted models while the significant covariates in the unadjusted models were run together in the adjusted model. The time to pubarche was 5% delayed among males than females (Time Ratio (TR) = 1.05: 95% CI: 1.03–1.05) in the unadjusted model. Males aged 20 to 24 years had delayed pubarche by 3% times than among those aged 15 to 19 years compared with 5% among females of the same age. In the adjusted model for the males (left panel of Table 4), significant differences were found in the TR to pubarche among the males' ages, the region of residence, rural versus urban and ethnicity. For instance, males aged 20 to 24 years had a 4% shorter time to pubarche (aTR = 0.96, 95% CI: 0.940.98) while controlling for other variables. Similarly, male residents in Northeast (aTR = 1.14, 95% CI: 1.07–1.21), in the Northwest (aTR = 1.20, 95% CI: 1.13–1.27) and in the Southwest (aTR = 1.18, 95% CI: 1.11–1.26) had delayed pubarche than males from the South East. Also, Yoruba males had delayed pubarche than Ibo males (aTR = 1.06, 95% CI: 1.01–1.12).

Females (right panel of Table 4) in the Southeast had the earlier time of pubarche than females from Northcentral (aTR = 1.17, 95% CI: 1.11–1.24), NortheEast (aTR = 1.20, 95% CI: 1.13–1.28), Northwest (aTR = 1.23, 95% CI: 1.16–1.31). SouthSouth (aTR = 1.13, 95% CI: 1.07–1.18), Southwest (aTR = 1.18, 95% CI: 1.11–1.25) while controlling for their ages, region of residence, place of residence, and ethnicity Other significant characteristics after controlling

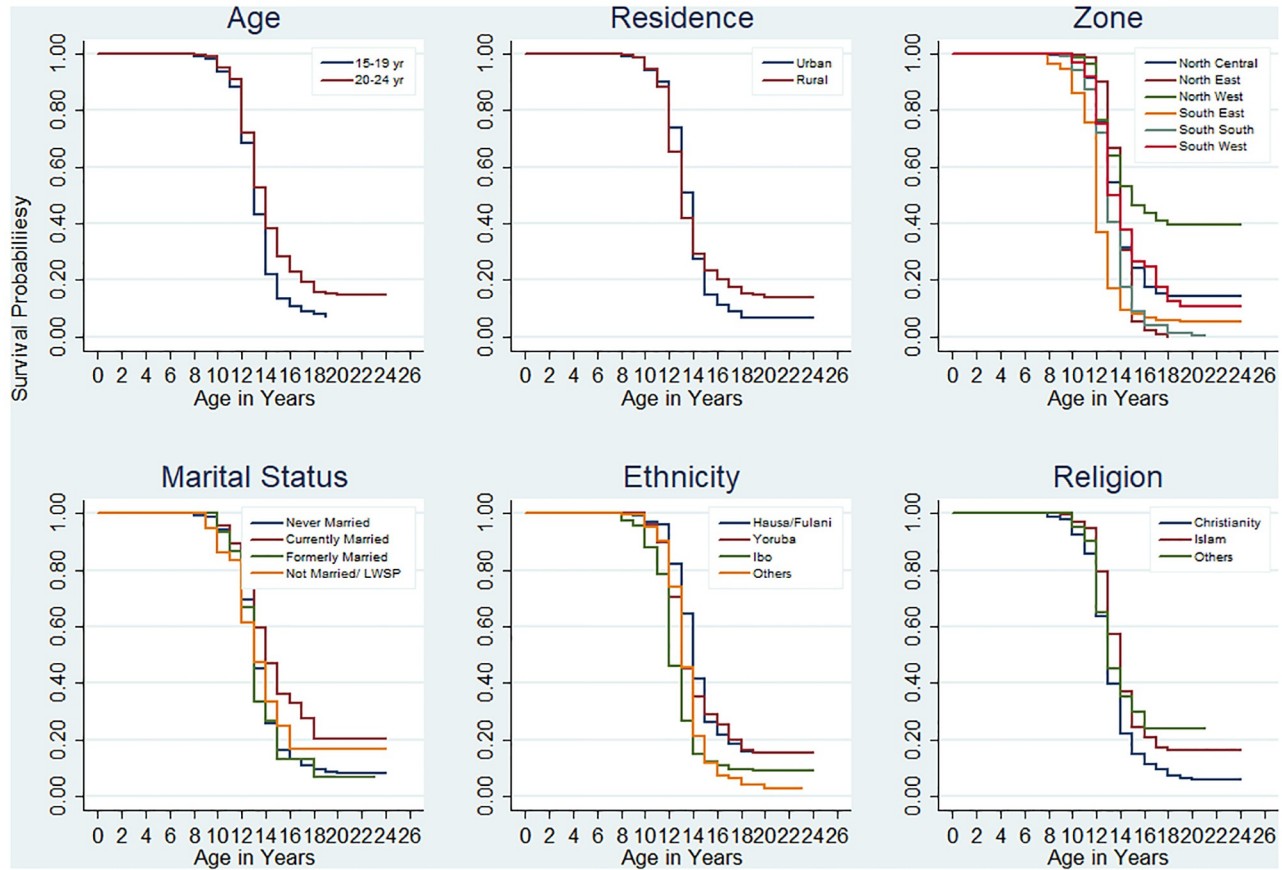

**Fig 2. Survivorship curves showing probabilities of developing pubic hair at different ages by selected characteristics of males.**

for other variables among the females was the respondent's household wealth category and age groups.

In summary, the age at the onset of pubarche among adolescents in Nigeria differed among males and females with earlier onset among females. Pubarche peaked among girls at 13 years and males at 14 years. The age at onset of pubarche differed by ethnicity, region and location of residence among males and differed by region and household wealth status among females. We observed a slightly delayed pubarche among older males and females. Pubarche was earlier among male adolescents in urban areas but not significantly different among rural and urban females.

## Discussion

This study was designed to assess the timing of pubarche retrospectively among male and female children in Nigeria to provide updated information on the timing of onset of pubic hair growth and to identify its prognostic factors. Recent reports showed that lack of information on pubarche is detrimental to the reproductive health of the youths as oftentimes, uninformed adolescents engaged in sexual risks behaviour which could lead to unwanted pregnancy, sexually transmitted diseases among others [5, 28, 33]. Many adolescents transit into adulthood without correct and adequate information on what puberty entails, the progression and how it affects their social and cognitive behaviours [1, 23]. We used survival analysis technique to provide updated information on the incidence and timing of puberty among male and female

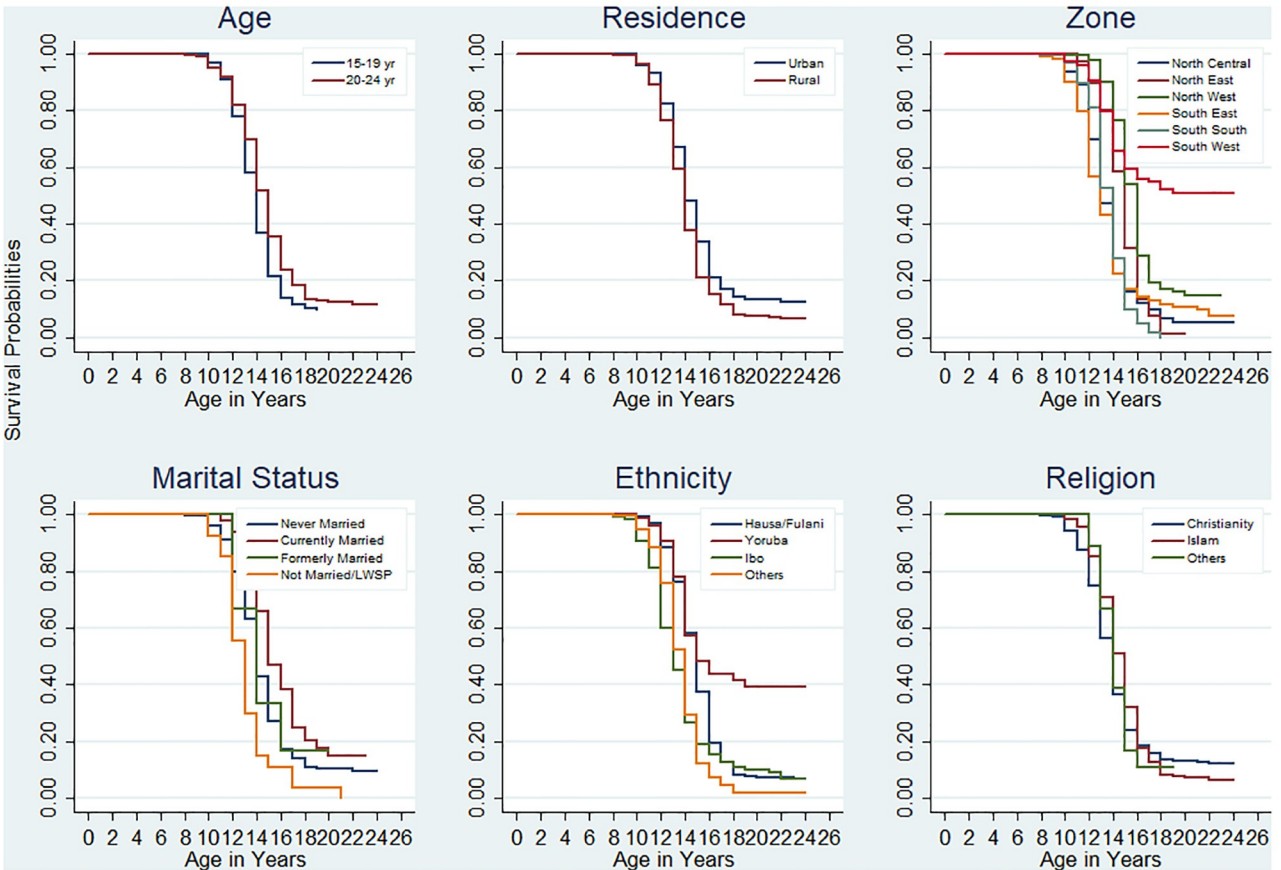

**Fig 3. Survivorship curves showing probabilities of developing pubic hair at different ages by selected characteristics of females.**

adolescents in Nigeria. We found gender divide in age at onset of pubarche and across the age cohort of the respondents. Other significant factors were the region of residence, place of residence, and ethnicity among the males and region of residence and household wealth status among the females.

Our analysis revealed that the mean age at pubarche among males aged 15 to 19 years and 20 to 24 years was 13.5 ($SD$ = 1.6) years and 14.2 ($SD$ = 2.2) years compared with 13.0 ($SD$ = 1.6) years and 13.5 ($SD$ = 2.1) years among females of the same age, respectively. Our estimates were comparable to the median age of attaining pubarche among Indian young population at 13.60 (95% CI: 13.3–14.0) years [34]. However, a much earlier gradual appearance of pubic hair in a 8-year, 7-month-old boy has been reported elsewhere [35]. While there was no significant rural-urban difference in age at pubarche among girls, we found earlier age at pubarche among rural males (13.7 years) than among the urban males (14.0 years). Our finding is at variance with the estimates reported in a Cubanian study which reported that pubarche was manifested at 11.18 years of age among rural male adolescents and 10.36 years among urban male adolescents [23]. A South African study also found an earlier age at pubic hair development among boys with an average decline of about one year (9.8 to 10.5 years) over ten years in the onset of pubarche [6]. We noted that our estimated mean age at pubarche among females at 13.2 years was higher than 11.3 years found among Iranian girls [25]. A multi-ethnic comparative study in the United Kingdom found that median age at pubarche occurred at earlier ages than in our study with 12.5 years among Bangladeshi girls, 11.6 years among British-

**Table 4. Unadjusted and adjusted determinants of risk of pubarche among adolescents and emerging adults.**

| | Males | | | | Females | | | |
|---|---|---|---|---|---|---|---|---|
| | Unadjusted Factors | | Adjusted Factors | | Unadjusted Factors | | Adjusted Factors | |
| | TR(95% CI) | p-value | aTR(95% CI) | p-value | TR(95%) CI | p-value | aTR(95%) CI | p-value |
| Sex (Female) | | | | | | | | |
| Male | **1.05(1.03–1.06)** | **<0.001** | | | | | | |
| Age (15–19 years) | | | | | | | | |
| 20–24 years | **1.03(1.01–1.06)** | **0.01** | **0.96(0.94–0.98)** | **<0.001** | **1.05(1.02–1.07)** | **<0.001** | **0.95(0.93–0.97)** | **<0.001** |
| Region (South East) | | | | | | | | |
| North Central | **1.04(1.01–1.08)** | **0.02** | 1.03(0.97–1.08) | 0.32 | **1.14(1.10–1.18)** | **<0.001** | **1.17(1.11–1.24)** | **<0.001** |
| North East | **1.15(1.11–1.18)** | **<0.001** | **1.14(1.07–1.21)** | **<0.001** | **1.18(1.14–1.23)** | **<0.001** | **1.20(1.13–1.28)** | **<0.001** |
| North West | **1.23(1.19–1.28)** | **<0.001** | **1.20(1.13–1.27)** | **<0.001** | **1.23(1.19–1.28)** | **<0.001** | **1.23(1.16–1.31)** | **<0.001** |
| South South | **1.05(1.02–1.08)** | **<0.001** | **1.05(1.01–1.10)** | **0.06** | **1.10(1.07–1.14)** | **<0.001** | **1.13(1.07–1.18)** | **<0.001** |
| South West | **1.24(1.19–1.29)** | **<0.001** | **1.18(1.11–1.26)** | **<0.001** | **1.16(1.12–1.21)** | **<0.001** | **1.18(1.11–1.25)** | **<0.001** |
| Residence (rural) | | | | | | | | |
| Urban | **1.04(1.02–1.06)** | **<0.001** | **1.03(1.01–1.05)** | **<0.001** | 1.00(0.98–1.02) | 0.93 | 1.01(0.97–1.02) | 0.71 |
| Education (None) | | | | | | | | |
| Pry/Quranic | 0.99(0.94–1.05) | 0.79 | | | 0.94(0.89–1.01) | 0.08 | | |
| Secondary | 0.98(0.93–1.03) | 0.46 | | | 0.96(0.90–1.01) | 0.14 | | |
| Higher | 1.01(0.95–1.07) | 0.84 | | | 0.95(0.89–1.02) | 0.16 | | |
| Ethnicity (Ibo) | | | | | | | | |
| Hausa/Fulani | **1.14(1.11–1.18)** | **<0.001** | 1.01(0.95–1.06) | 0.86 | **1.16(1.12–1.21)** | **<0.001** | 0.99(0.94–1.04) | 0.72 |
| Yoruba | **1.19(1.15–1.23)** | **<0.001** | **1.06(1.01–1.12)** | **0.03** | **1.12(1.08–1.16)** | **<0.001** | 0.98(0.93–1.03) | 0.46 |
| Others | **1.04(1.01–1.07)** | **0.01** | 0.99(0.95–1.04) | 0.81 | **1.09(1.06–1.13)** | **<0.001** | 0.98(0.93–1.03) | 0.34 |
| Wealth Category (Lowest) | | | | | | | | |
| Average | 0.99(0.97–1.02) | 0.47 | | | **1.04(1.02–1.07)** | **0.04** | 1.01(0.96–1.03) | 0.06 |
| Highest | 1.01(0.99–1.04) | 0.29 | | | **1.05(1.03–1.08)** | **0.03** | 1.02(1.01–1.05) | 0.04 |
| _cons | | | 12.02(11.2–12.8) | <0.001 | | | 13.01(12.73–13.3) | <0.001 |
| /ln_sig* | | | -1.8(-1.84–1.75) | <0.001 | | | -1.71(-1.75–1.65) | <0.001 |
| /kappa* | | | -0.47(-0.64–0.3) | <0.001 | | | -0.74(-0.88–0.61) | <0.001 |
| Sigma* | | | **0.17(0.16–0.17)** | **<0.001** | | | **0.18(0.17–0.19)** | **<0.001** |

TR Time Ratio aTR adjusted Time Ratio LWSP Living with Sexual Partner significant values are printed in bold *Parameters estimates of Generalized Gamma Model

Bangladeshi and 10.9 years for white British [36]. Similarly, lower pubarche age of 10.25 years was found among African American Non- Hispanic boys, 11.43 years among Hispanic boys and 11.47 years among Non-Hispanic White boys in the United States [36, 37] The younger (15 to 19 years) respondents in our study had delayed commencement of pubarche compared with those aged 20–24 years. This finding suggests an overall lower age at onset of pubarche over time. We observed later maturing among boys compared to girls. This is consistent with earlier studies [38]. An earlier study in Nigeria has also reported that more males, 14.2% significantly had a late rate of pubertal development than the females 5.1% [27].

Nonetheless, we found that a significant proportion of the respondents (12.1% among males and 1.7% among females) had not attained pubarche. This signifies that the age at pubarche is much earlier among females than the males. This was more prominent among females aged 20 to 24 years, males in the South-West region, females in the North-West region and Yoruba ethnic males. The differentials along ethnic divides could be attributed to differences in nutrition as well as possible genetic differentials.

Furthermore, the socio-economic class was found to be significantly associated with the timing of the onset of pubarche among only the girls in our study. Better wealth categories of households increased their risk of early pubarche. This finding was corroborated in other research that found socioeconomic class to be a significant predictor of attainment of puberty among Nigerian adolescents [27]. This finding could be linked to access to better nutritional intakes among urban girls [15, 20–22]. A birth cohort study from Australia found that boys from socially disadvantaged households had a fourfold increase in the rate of early puberty while girls had the nearly twofold early onset of puberty [39], although the mechanism through which this relationship exists was not established. The differentials we found in age at pubarche viz-a-viz wealth quintile gave credence to the fact that nutrition and dietary intake could significantly impact early age at pubarche since children from wealthier homes are more likely to have balanced diets than others [40–42].

There were notable differences in the time of the event of pubarche among male and female respondents in different regions of the country. It is striking that a quarter of males in North Central experienced pubarche at age 12 which is earlier than 13 years for girls in the same region. Although the association between rural-urban differentials in residence and timing of pubarche was significant among males, it was not significant among the females. However, there was a slight difference in pubarche timing among the females whereby about half of the females in rural areas would have experienced pubarche earlier than those in urban areas. Contrary findings were reported in a recent study in Nigeria wherein female adolescents residing in urban areas had a higher probability of earlier pubarche compared to those residing in rural areas [8]. Our finding, however, corroborated an earlier Cubanian study that found significant higher (11,18 years) age at the onset of pubarche in the rural areas than in the urban areas (10,36 years) [23]. Also, we found differences in onset of pubarche along ethnic divides with earlier occurrences among Yoruba and Hausa/Fulani males and females than their Igbo counterparts. This finding is in consonance which reported ethnic differences have identified in maturation among males and females [1].

Another significant outcome that the timing of pubarche differed across the geographical zones in Nigeria especially among females. Timing of pubarche was earlier among males in the Northwest, Northeast and Southwest than in the Southeast. Also, the females in the other 5 regions in Nigeria had delayed pubarche than girls in Southeast. Our finding is consistent with the literature. For instance, significant variabilities were found in the timing of pubarche across the eleven regions in Iran [25]. The authors ascribed their findings to differences in dietary intakes.

Theoretically, the ecological model [26] which was adopted in this study is appropriate as we observe a complex interplay among individuals, relationships, community and societal factors in the timing of pubarche. The theoretical model was used to link the diverse and complex range of characteristics such as parental and peer relationships, as well as the broader culture, in shaping adolescent development, including pubarche. We identified a pathway of association between individual, household community and societal characteristics and the onset of pubarche. Our findings align with existing discourse on the age at pubarche. We found that the differences in timings of onset of pubarche can be attributed to biological and socio-demographic variabilities. This is in tandem with literature [1, 8, 15]. The findings affirmed that association exist between socioeconomic status, education and health outcomes and age at pubarche [19]. However, our study fell short of establishing the theoretical belief that age at pubarche is highly impacted by nutritional intake as alluded to in the literature [20–22]. Our findings also corroborated the literature on the role of race and ethnicity on age at onset of pubic hair [23].

## Strength and limitations

The estimates in this study were based on data from a nationally representative sample. This has made our findings reproducible and generalizable across Nigeria. We used self-reported ages at pubarche reported by the participants without any means of external validation. The accuracy of ages at pubarche might have been subjected to recall bias. We used other questions to try to validate responses; for example, we had about 20% of the respondents either couldn't remember the exact date or provided invalid ages that were not consistent with other responses, and we deleted those participants from the analysis sample. The prognostic factors assessed in this study were limited to sociodemographic and behavioural characteristics, whereas some clinical such as adrenal insufficiency, panhypopituitarism and hypothyroidism may influence timing of pubarche [43, 44]. Use of cross-validation questions, the representativeness, and large sample size used in the study would have minimized the effect of such bias on our estimates.

## Recommendations

Our study has provided relevant information and updated the body of knowledge on the timing of pubarche and the associated factors among boys and girls in Nigeria. This information can be used to guide adolescents, parents, guardians, and the community at large in making adequate preparations for this pivotal development in children. The outcomes can be used to reduce anxieties among girls, mothers and guardians and guide health care providers in implementing appropriate health programmes. Our study has public health and social implications. Parents and guardians should take note of general reduction in time to pubarche. It is expedient to create awareness and enlightenment among parents and guardians on the timing of pubarche and the need to make adequate preparation for it to offer appropriate guides and supports to adolescents. The knowledge of the age of onset of pubarche is of great importance for parents regarding the appropriate time to start conversations with their children about sex and sexuality. Also, it is important to incorporate reproductive health education into the school curriculum in the country to assist adolescents as they transit into young adults. Such reproductive health education should encompass health education and promotion on fertility, sexual abuse and violence, body formation, HIV transmission and prevention. The reported ethnic differences in age at pubarche may be related to myths about diet (its richness or its absence), based on the results of wealth quintiles and residential location on pubarche to enable parents and policy makers to build better understandings of pubarche for the young adolescents. A qualitative study may be necessary to explore the form of recognising and correcting myths surrounding nutrition and pubarche etc. Policymakers, researchers, reproductive and youth programme officers and all stakeholders should consider the timing of pubarche in planning and development of adolescent health programmes. We also recommend further studies on clinical characteristics that may explain the timing of pubarche.

## Acknowledgments

We acknowledged the support received by the first author from the Consortium for Advanced Research Training in Africa (CARTA). CARTA is jointly led by the African Population and Health Research Center and the University of the Witwatersrand and funded by Carnegie Corporation of New York (Grant No. G-19-57145), Sida (Grant No:54100113), Uppsala Monitoring Center, Norwegian Agency for Development Cooperation (Norad), and by the Wellcome Trust [reference no. 107768/Z/15/Z] and the UK Foreign, Commonwealth & Development Office, with support from the Developing Excellence in Leadership, Training and Science in Africa (DELTAS Africa) programme. The statements made and views expressed are solely the

responsibility of the Fellow. For the purpose of open access, the author has applied a CC BY public copyright license to any Author Accepted Manuscript version arising from this submission.

## Author Contributions

**Conceptualization:** Adeniyi Francis Fagbamigbe, Mary Obiyan, Olufunmilayo I. Fawole.

**Data curation:** Adeniyi Francis Fagbamigbe.

**Formal analysis:** Adeniyi Francis Fagbamigbe.

**Investigation:** Adeniyi Francis Fagbamigbe.

**Methodology:** Adeniyi Francis Fagbamigbe, Olufunmilayo I. Fawole.

**Project administration:** Adeniyi Francis Fagbamigbe, Mary Obiyan.

**Resources:** Adeniyi Francis Fagbamigbe, Mary Obiyan.

**Supervision:** Olufunmilayo I. Fawole.

**Visualization:** Adeniyi Francis Fagbamigbe.

**Writing – original draft:** Adeniyi Francis Fagbamigbe, Mary Obiyan, Olufunmilayo I. Fawole.

**Writing – review & editing:** Adeniyi Francis Fagbamigbe, Mary Obiyan, Olufunmilayo I. Fawole.

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
