## [Editor Report · Decision Letter 0]

30 Mar 2022

PONE-D-22-01702Gender differentials in the timing and prognostic factors of pubarche in Nigeria: Application of Generalized gamma survival modelPLOS ONE

Dear Dr. Fagbamigbe,

Thank you for submitting your manuscript to PLOS ONE. After careful consideration, we feel that it has merit but does not fully meet PLOS ONE’s publication criteria as it currently stands. Therefore, we invite you to submit a revised version of the manuscript that addresses the points raised during the review process.

We look forward to receiving your revised manuscript.

Kind regards,

Jong In Kim

Academic Editor

PLOS ONE

Journal Requirements:

2.Thank you for stating in your Funding Statement: 

This research was supported by the Consortium for Advanced Research Training in Africa (CARTA). CARTA is jointly led by the African Population and Health Research Center and the University of the Witwatersrand and funded by the Wellcome Trust (UK) (Grant No: 087547/Z/08/Z), the Carnegie Corporation of New York (Grant No--B 8606.R02), Sida (Grant No:54100029)”. CARTA had trained the first author in research methodologies. The statements made and views expressed are solely the responsibility of the author.)

When you resubmit, please ensure that you provide the correct grant numbers for the awards you received for your study in the ‘Funding 4 section.

Additional Editor Comments:

This paper assessed the timing of pubarche among adolescents and young adults in Nigeria and identified prognostic factors of the timing by obtaining information on youths' sexual and reproductive developments in a population survey among in-school and out-of-school youths aged 15 to 24 years in Nigeria. A total of 1174 boys and 1004 girls provided valid information on pubarche.

Please supplement the following.

1. Supplement the theoretical aspects of the discussion.

2. Present the overall frame graphic.

---

## [Author Response · Author response to Decision Letter 0]

16 Jun 2022

Respected Editor,

Thank you for your letter and the opportunity to revise manuscript entitled “Timing and Prognostic Factors of Pubarche in Nigeria: Parametric Survival Analysis”. 

The suggestions offered by the reviewers have been immensely helpful, and we also appreciate your insightful comments on revising the abstract and other aspects of the paper.

I hereby include the reviewer comments below and provided a point-by-point to all the comments of the Editor and the reviewers

The revisions were approved by all the authors. The manuscript has been scrutinized for requirement in the author instructions, as well. We hope the revised manuscript will better suit the Journal.

Thank you

Adeniyi Fagbamigbe

Journal Requirements:

Thank you. We have corrected all the requirements

2.Thank you for stating in your Funding Statement: 

This research was supported by the Consortium for Advanced Research Training in Africa (CARTA). CARTA is jointly led by the African Population and Health Research Center and the University of the Witwatersrand and funded by the Wellcome Trust (UK) (Grant No: 087547/Z/08/Z), the Carnegie Corporation of New York (Grant No--B 8606.R02), Sida (Grant No:54100029)”. CARTA had trained the first author in research methodologies. The statements made and views expressed are solely the responsibility of the author.)

Thank you. We have revised the statements L456. It was a support not funding per sai

When you resubmit, please ensure that you provide the correct grant numbers for the awards you received for your study in the ‘Funding 4 section.

Thank you. We have revised the statements

Thank you. We have revised the statements L456- 465

Thank you. We have revised the references and updated the links. We used the Mendeley software. All our references were cited and listed at the end. 

We retracted (CIA World Factbook, 2017) Ref 29 because same information was in reference 30 

Additional Editor Comments:

This paper assessed the timing of pubarche among adolescents and young adults in Nigeria and identified prognostic factors of the timing by obtaining information on youths' sexual and reproductive developments in a population survey among in-school and out-of-school youths aged 15 to 24 years in Nigeria. A total of 1174 boys and 1004 girls provided valid information on pubarche.

Please supplement the following.

Thank you

1. Supplement the theoretical aspects of the discussion.

We have supplemented the theoretical aspects viz-a-viz our findings. Kindly see L403-L411

2. Present the overall frame graphic.

We have provided the overall frame graphic L95-L98

---

## [Decision Letter · Decision Letter 1]

22 Sep 2022

PONE-D-22-01702R1Gender differentials in the timing and prognostic factors of pubarche in Nigeria: Application of Generalized gamma survival modelPLOS ONE

Dear Dr. Fagbamigbe,

Thank you for submitting your manuscript to PLOS ONE. After careful consideration, we feel that it has merit but does not fully meet PLOS ONE’s publication criteria as it currently stands. Therefore, we invite you to submit a revised version of the manuscript that addresses the points raised during the review process.

Two reviewers have analyzed the manuscript. Reviewer 2, in particular, notes issues that require clarification / action regarding the methodology, the statistical analysis and the reporting.

We look forward to receiving your revised manuscript.

Kind regards,

José Antonio Ortega, Ph.D.

Academic Editor

PLOS ONE

Reviewers' comments:

Reviewer's Responses to Questions

**Comments to the Author**

1. If the authors have adequately addressed your comments raised in a previous round of review and you feel that this manuscript is now acceptable for publication, you may indicate that here to bypass the “Comments to the Author” section, enter your conflict of interest statement in the “Confidential to Editor” section, and submit your "Accept" recommendation.

Reviewer #1: All comments have been addressed

Reviewer #2: (No Response)

2. Is the manuscript technically sound, and do the data support the conclusions?

Reviewer #1: Yes

Reviewer #2: No

3. Has the statistical analysis been performed appropriately and rigorously? 

Reviewer #1: Yes

Reviewer #2: No

4. Have the authors made all data underlying the findings in their manuscript fully available?

Reviewer #1: Yes

Reviewer #2: Yes

5. Is the manuscript presented in an intelligible fashion and written in standard English?

Reviewer #1: Yes

Reviewer #2: No

6. Review Comments to the Author

Reviewer #1: All are included in the comment. The authors did a rigorous analysis which is relevant to the objective of the study.

Reviewer #2: 1. It may be more appropriate to put the study design in the title rather than the statistical technique used.

2. Abstract, “median survival time to pubarche …” is misleading and better remove the word ‘survival’. It’s time-to-event analysis not necessarily survival.

3. Abstract, “Every additional one-year increase in the ages of both males and females reduces the risk of pubarche by 1%”, this statement is not clear to me. I thought pubarche is attainment of pubic hair. So, increasing age the risk of pubarche should increase, not decrease. That’s also their finding in this study.

4. Abstract, conclusion: the authors conclude that pubarche time varied significantly by region, ethnicity, and location of residence. However, such information was not provided at all in the results section of the abstract. The conclusion in the abstract should reflect only the results given in the abstract.

5. The conclusion that girls attain pubarche before boys is more or less an established fact. Rather, it would be more interesting if the authors found that the median age for pubarche among their study population is significantly different from the expected ranges as their hypothesis was “the timing of pubarche in Nigeria differs from other countries” (stated in the abstract). There appears marked discrepancy between the study question and answer to the question.

6. Introduction, “Besides, the available ones are dated”. What does this statement mean? it’s not clear to me.

7. Study design: my understanding is that the authors took a random sample of the population and conducted a survey. I didn’t understand how this could be labeled retrospective study unless I am missing something. This rather looks like a cross-sectional survey with a prospective data collection. If the data collection was from already stored data sources and the authors didn’t survey the study participants, this needs to be clearly stated.

8. The authors used multi-stage cluster sampling which is different from simple random sampling. Multi-stage cluster sampling requires correction for the design effect (the ratio of the variance from the cluster sampling to that expected from simple random sampling). This wasn’t the case in their study. Furthermore, they calculated and surveyed 2952 subjects but the data presented was only for 2178 participants. They excluded 774 participants which underpowers the study. As this was a cross-sectional study, I wonder why the authors didn’t continue to interview participants until their sample size was achieved to replace those who were excluded.

9. The statistical analysis section is extremely redundant but yet incomplete. The paper is about time-to-pubarche and not review of the statistical technique itself. They don’t need to discuss the survival function and hazard function in such detail; it’s enough if they just state these were the methods used to analyze the data. Trying to give unnecessary details, they have missed mentioning other simple statistical methods used in their study.

10. The authors mention that there was a delay in pubarche among older males and females. Could this be simple recall bias as the older ones are years away from the age of pubarche at the time of the interview?

11. Recommendation section: the authors recommend regular, repetitive, and comprehensive school-based sexuality education. Conclusions and recommendations should be based on the results of the study.

7. PLOS authors have the option to publish the peer review history of their article (what does this mean?). If published, this will include your full peer review and any attached files.

Reviewer #1: No

Reviewer #2: **Yes: **Endale Tefera

---

## [Author Response · Author response to Decision Letter 1]

2 Oct 2022

Dear Editor.

We appreciate your efforts and our eminent reviewers. We have now completed the revision and provided point-by-point response.

Thank you

Reviewer 1

Gender differentials in the timing and prognostic factors of pubarche in Nigeria: Generalized gamma survival model

Thank you for the opportunity to review the manuscript with the above title. The paper is well written. There are a few things that need to be addressed. My concerns are written in green. 

Abstract

Paucity of data exists on the timing of puberty, particularly the pubarche, in developing countries, which has hitherto limited the knowledge of the timing of pubarche, and assistance offered by physicians to anxious young people in Nigeria. I think paucity 

THIS IS NOT CLEAR- YOU THINK PAUCITY?

We hypothesized that the timing of pubarche in Nigeria differs from other countries.

It appears that there is a comparative study between Nigeria and other developing countries. It is better to hypothesize within the geographical divides in Nigeria. Remove other countries. Also, the first sentence should focus on Nigeria. 

THANK YOU, WE HAVE CHANGED THIS

Our findings will aid medical practitioners in providing appropriate advice and support on pubarche-related issues among adolescents in Nigeria.

One would like to see how the evidence of pubarche anxiousness in relation to request for medical assistance. 

THANK YOU, WE HAVE ADDED THIS

Introduction 

The need to update knowledge and to inform surrounding adults to ease anxieties that often comes with the unpredictability of age at pubarche necessitated this study.

Is there evidence of the needs and anxieties with regard to pubarche? 

THANK YOU, WE PROVIDED REFERENCES 1& 5

The knowledge of risk factors associated with pubarche could help medical practitioners provide evidence-based care and guidance to adolescents and their guardians.

I expected to see evidence of seeking medical assistance in relation to pubarche issues.

UNFORTUNATELY, WE DON’T HAVE A JOURNAL REPORTING THESE, HOWEVER, IT REMAINS AN UNPUBLISHED ISSUE

 We hypothesized that better educational attainment could improve decisions on healthy food that may, in turn, affect the onset of pubarche; pubertal timing has been demonstrated to be a function of nutritional intake during childhood. 

The association between place of residence and timing of pubarche can be explained by the differentials in nutritional intake in rural and urban areas. Also, an Indian study reported rural-urban differences in age at pubarche 23. Can the authors elaborate on this study? Was nutrition implicated? 

YES TO AN EXTENT, WE CITED REFERENCE 15 AS SHOWN BELOW

“NUTRITIONAL INTAKES HAVE BEEN REPORTED TO BE DIFFERENT AMONG RURAL AND URBAN CHILDREN, SINCE BOTH HAVE DIFFERENTIAL ACCESS TO DIFFERENT TYPES OF FOODS 15.”

Race and ethnicity have been identified as associated factors with age at pubarche 24. This was ascribed to differences in the levels of metabolism and insulin across different races. This does not seem to be complete. 

THANK YOU, WE HAVE MADE SENSE OUT OF THE SENTENCE

Thus, we hypothesized that the timing of pubarche in Nigeria differs from other countries and that age of pubarche in the country has changed over time compared with what is available in literature.

Can this hypothesis be based on what is found in Nigeria? Mentioning other countries that are not part of the study will make testing the hypothesis impossible. 

WE TOTALLY AGREE. THIS HAS BEEN CORRECTED

Significance of the study

Recent reports showed that the lack of information on puberty is detrimental to the reproductive health of the youths. Where is the evidence of this? 

WE HAVE PROVIDED REFERENCE NUMBER 4

Many adolescents transition to adulthood without correct and adequate information on what puberty entails and how it affects their social and cognitive behaviours. This does not seem to belong here.

THANK YOU. WE HAVE MOVED IT TO 

Methods: This section is acceptable 

Study Design

The study design was a retrospective nationally representative population survey. What is the name of the survey and the year it was conducted? 

Thank you. We have provided this as 

The data used for this analysis was collected during the “Youths and adolescents sexual and reproductive health survey” in 2017.

Line 54-56: Although different studies have estimated the timing of pubarche and identified its associated risk factors in most regions of the world 6–10, such studies are scarce for the Nigeria setting. Besides, the available ones are dated 11,12. Do you mean it is outdated? 

YES WE HAVE CORRECTED THIS

The Theoretical / Conceptual Framework is appropriate and well-articulated. 

The analysis is very informative and well done.

THANK YOU

Discussion

 We used survival analysis technique to provide updated information on the incidence and timing of puberty among male and female adolescents in Nigeria. This is for the methodology section. It seems that results are being repeated in the discussion. 

NOT REALY, WE ONLY USED THAT SENETENCE AS INTRODUCTION TO REFRESH THE READERS

Reviewer #1: All are included in the comment. The authors did a rigorous analysis which is relevant to the objective of the study.

THANK YOU.

Reviewer #2: 

Reviewer #2: 1. It may be more appropriate to put the study design in the title rather than the statistical technique used.

THANK YOU. WE HAVE CORRECTED THIS

2. Abstract, “median survival time to pubarche …” is misleading and better remove the word ‘survival’. It’s time-to-event analysis not necessarily survival.

THANK YOU. WE HAVE CORRECTED THIS

3. Abstract, “Every additional one-year increase in the ages of both males and females reduces the risk of pubarche by 1%”, this statement is not clear to me. I thought pubarche is attainment of pubic hair. So, increasing age the risk of pubarche should increase, not decrease. That’s also their finding in this study.

THANK YOU. WE HAVE CORRECTED THIS

The statement read “Every additional one-year in the ages of both males and females increases the risk of pubarche by 1%.”

4. Abstract, conclusion: the authors conclude that pubarche time varied significantly by region, ethnicity, and location of residence. However, such information was not provided at all in the results section of the abstract. The conclusion in the abstract should reflect only the results given in the abstract.

THANK YOU. WE AGREED TOTALLY AND HAVE PROVIDE SOME OF THESE BUT WE ARE LIMITED BY SPACE.

5. The conclusion that girls attain pubarche before boys is more or less an established fact. Rather, it would be more interesting if the authors found that the median age for pubarche among their study population is significantly different from the expected ranges as their hypothesis was “the timing of pubarche in Nigeria differs from other countries” (stated in the abstract). There appears marked discrepancy between the study question and answer to the question.

THANK YOU. THE OTHER REVIEWERS HAD SUGGESTED THAT WE CHANGE THE RESEARCH QUESTIONS TO “We hypothesized that the timing of pubarche in Nigeria differs by geographical regions and other characteristics”, which we obliged. Nonetheless, we have compared our findings with available data elsewhere in the discussions.

6. Introduction, “Besides, the available ones are dated”. What does this statement mean? it’s not clear to me.

THANK YOU, IT IS “outdated” 

YES WE HAVE CORRECTED THIS

7. Study design: my understanding is that the authors took a random sample of the population and conducted a survey. I didn’t understand how this could be labeled retrospective study unless I am missing something. This rather looks like a cross-sectional survey with a prospective data collection. If the data collection was from already stored data sources and the authors didn’t survey the study participants, this needs to be clearly stated.

THANK YOU. IT WAS CROSS-SECTIONAL. WE USED THE TERM RETROSZPECTIVE TO INDICTE THAT THE RESPONDENTS HAD TO LOOK BACK INTO PAST EVENTS. A TIME TO EVENT DATA COULD EITHER BE RETROSPECTIVE OR PROSPECTIVE. OURS IS RETROSPECTIVE. 

WE HAVE REMOVED THE TERM RETROSPECTIVE SO AS IT AVOID CONFUSION.

8. The authors used multi-stage cluster sampling which is different from simple random sampling. Multi-stage cluster sampling requires correction for the design effect (the ratio of the variance from the cluster sampling to that expected from simple random sampling). This wasn’t the case in their study. Furthermore, they calculated and surveyed 2952 subjects but the data presented was only for 2178 participants. They excluded 774 participants which underpowers the study. As this was a cross-sectional study, I wonder why the authors didn’t continue to interview participants until their sample size was achieved to replace those who were excluded.

MANY THANKS FOR THIS COMMENT. THE EXPECTED SAMPLE SIZE FOR SIMPLE RANDOM SAMPLING WAS 328 AFTER PROVISION FOR 10% NON-RESPONSE. WE APPLIED A DESIGN EFFECT OF 1.5 BEEN A CLUSTER SAMPLING TO ARRIVE AT 492 FOR EACH OF THE SIX ZONES (492*6=2952). 

THIS PAPER IS A PART OF MULTI-OBJECTIVE SURVEY, WHEREIN INVALID RESPONSES FOR AGE AT PUBARCHES FOR THIS MANUSCRIPT WERE ONLY DETECTED AT ANALYSIS LEVEL. DESPITE THIS, THE POWER WAS 0.9875 (98.8%) WHICH WAS CLEARLY ABOVE THE NORMAL THRESHOLD OF 80%

9. The statistical analysis section is extremely redundant but yet incomplete. The paper is about time-to-pubarche and not review of the statistical technique itself. They don’t need to discuss the survival function and hazard function in such detail; it’s enough if they just state these were the methods used to analyze the data. Trying to give unnecessary details, they have missed mentioning other simple statistical methods used in their study.

WE HAVE REMOVED REDUNDANCY BUT RETAINED JUSTIFICATION FOR USE OF GAMMA MODELS. THE OTHER STATISTICAL METHODS WERE MENTIONED IN THE PARAGRAPH BEFORE “ANALYSIS OF TIME TO EVENT”

10. The authors mention that there was a delay in pubarche among older males and females. Could this be simple recall bias as the older ones are years away from the age of pubarche at the time of the interview?

THANK YOU, RECALL BIAS CANNOT BE RULED OUT DESPITE EFFORTS TO MINIMISE THIS AT DATA COLLECTION STAGE. WE HAVE STATED THIS AS A STUDY LIMITATION

11. Recommendation section: the authors recommend regular, repetitive, and comprehensive school-based sexuality education. Conclusions and recommendations should be based on the results of the study.

THANK YOU, WE HAVE REMOVED THIS

---

## [Decision Letter · Decision Letter 2]

25 Oct 2022

PONE-D-22-01702R2Gender differentials in the timing and prognostic factors of pubarche in NigeriaPLOS ONE

Dear Dr. Fagbamigbe,

Thank you for submitting your manuscript to PLOS ONE. After careful consideration, we feel that it has merit but does not fully meet PLOS ONE’s publication criteria as it currently stands. Therefore, we invite you to submit a revised version of the manuscript that addresses the points raised during the review process. Reviewer 1 was satisfied with the previous version and reviewer 2 feels that their comments have been addressed in this revision. My main concern is with the data availability. You are saying that the data is available without restrictions but it is not included with the manuscript and there is no link provided to the data location. Also, a google search for the name of the survey did not provide any meaningful link. If you were not involved in data production, it would also help if you provide cite/link to published materials (eg, survey report). As you know authors are required to make all data underlying the findings described fully available, without restriction, and from the time of publication. PLOS allows rare exceptions to address legal and ethical concerns. See the PLOS Data Policy and FAQ for detailed information.

We look forward to receiving your revised manuscript.

Kind regards,

José Antonio Ortega, Ph.D.

Academic Editor

PLOS ONE

Journal Requirements:

Reviewers' comments:

Reviewer's Responses to Questions

**Comments to the Author**

1. If the authors have adequately addressed your comments raised in a previous round of review and you feel that this manuscript is now acceptable for publication, you may indicate that here to bypass the “Comments to the Author” section, enter your conflict of interest statement in the “Confidential to Editor” section, and submit your "Accept" recommendation.

Reviewer #2: All comments have been addressed

2. Is the manuscript technically sound, and do the data support the conclusions?

Reviewer #2: Yes

3. Has the statistical analysis been performed appropriately and rigorously? 

Reviewer #2: Yes

4. Have the authors made all data underlying the findings in their manuscript fully available?

Reviewer #2: Yes

5. Is the manuscript presented in an intelligible fashion and written in standard English?

Reviewer #2: Yes

6. Review Comments to the Author

Reviewer #2: (No Response)

7. PLOS authors have the option to publish the peer review history of their article (what does this mean?). If published, this will include your full peer review and any attached files.

Reviewer #2: **Yes: **Endale Tefera

---

## [Author Response · Author response to Decision Letter 2]

27 Oct 2022

My main concern is with the data availability. You are saying that the data is available without restrictions but it is not included with the manuscript and there is no link provided to the data location. Also, a google search for the name of the survey did not provide any meaningful link. If you were not involved in data production, it would also help if you provide cite/link to published materials (eg, survey report). As you know authors are required to make all data underlying the findings described fully available, without restriction, and from the time of publication. PLOS allows rare exceptions to address legal and ethical concerns. See the PLOS Data Policy and FAQ for detailed information.

The data link has been provided

---

## [Editor Report · Decision Letter 3]

28 Oct 2022

PONE-D-22-01702R3Gender differentials in the timing and prognostic factors of pubarche in NigeriaPLOS ONE

Dear Dr. Fagbamigbe,

Thank you for submitting your manuscript to PLOS ONE. After careful consideration, we feel that it has merit but does not fully meet PLOS ONE’s publication criteria as it currently stands. Therefore, we invite you to submit a revised version of the manuscript that addresses the points raised during the review process.

Thank you for providing the link to the data. It requires a minimal change, though, since the data is not fully anonymized. A variable called "Name" is available which could be used to locate the interviewees, something particularly delicate given the topics covered. Please remove the name column or/and replace it with an arbitrary id number.

We look forward to receiving your revised manuscript.

Kind regards,

José Antonio Ortega, Ph.D.

Academic Editor

PLOS ONE
---

## [Author Response · Author response to Decision Letter 3]

2 Nov 2022

The link has been provided

https://drive.google.com/file/d/1BSeAN7mG4vOYICD7Wec4UvaSYKUhwoNo/view?usp=share_link

---

## [Editor Report · Decision Letter 4]

4 Nov 2022

Gender differentials in the timing and prognostic factors of pubarche in Nigeria

PONE-D-22-01702R4

Dear Dr. Fagbamigbe,

We’re pleased to inform you that your manuscript has been judged scientifically suitable for publication and will be formally accepted for publication once it meets all outstanding technical requirements.

Kind regards,

José Antonio Ortega, Ph.D.

Academic Editor

PLOS ONE

Additional Editor Comments (optional):

The updated data looks complete and is anonymized. The submission is therefore ready for publication.
---

## [Editor Report · Acceptance letter]

11 Nov 2022

PONE-D-22-01702R4 

Gender differentials in the timing and prognostic factors of pubarche in Nigeria 

Dear Dr. Fagbamigbe:

I'm pleased to inform you that your manuscript has been deemed suitable for publication in PLOS ONE. Congratulations! Your manuscript is now with our production department. 

Kind regards, 

on behalf of

Dr. José Antonio Ortega 

Academic Editor

PLOS ONE